# Two-Dimensional Barrage Jamming against SAR Using a Frequency Diverse Array Jammer

**DOI:** 10.3390/s23052449

**Published:** 2023-02-22

**Authors:** Jingke Zhang, Yonghu Zeng, Zongfeng Qi, Liandong Wang, Ya Wang, Xujian Shen

**Affiliations:** State Key Laboratory of Complex Electromagnetic Environment Effects on Electronics and Information System, Luoyang 471000, China

**Keywords:** synthetic aperture radar (SAR), frequency diverse array (FDA), frequency offset, micro-motion modulation, barrage jamming

## Abstract

Due to the modulation of tiny frequency offset on the array elements, a frequency diverse array (FDA) jammer can generate multiple range-dimension point false targets, and many deception jamming methods against SAR using an FDA jammer have been studied. However, the potential of the FDA jammer to generate barrage jamming has rarely been reported. In this paper, a barrage jamming method against SAR using an FDA jammer is proposed. To achieve two-dimensional (2-D) barrage effect, the stepped frequency offset of FDA is introduced to generate range-dimensional barrage patches, and the micro-motion modulation is employed to increase the extent of barrage patches along the azimuth direction. Mathematical derivations and simulation results demonstrate the validity of the proposed method in generating flexible and controllable barrage jamming.

## 1. Introduction

Due to the capacity involved in providing high resolution images independent of sunlight illumination and weather conditions, synthetic aperture radar (SAR) has been widely used in both military and civil applications, such as target detection and recognition [1,2,3]. Meanwhile, to avoid the target being detected and observed by the SAR, the electronic countermeasures (ECM) against SAR have been developed rapidly in recent decades [4,5,6]. Generally, from the point of the view of jamming effect, SAR jamming can be divided into barrage jamming and deceptive jamming [7,8,9,10]. The purpose of barrage jamming is to form suppression patches or stripes in the SAR image to mask the real target by transmitting a high-power noise-like signal or modulated coherent jamming signal [11,12,13,14], while deceptive jamming is used to modulate the scattering characteristics of the real target on the intercepted radar signal to form false targets with high fidelity [15,16,17,18]. Various jamming techniques of both types have been developed, which are always realized by a jammer employed with a traditional single-channel antenna or a phased-array (PA) antenna.

Recently, an emerging technique called frequency diverse array (FDA) antenna has attracted considerable attention in the both the radar and electronic countermeasures fields [19,20,21,22,23]. Unlike the PA antenna, the FDA antenna modulates signals with a frequency offset much smaller than the carrier frequency across its array elements to produce a joint range-angle-time dependent transmit beam pattern, which will produce multiple range-dimensional point false targets when the FDA antenna is employed by a jammer [21]. Due to its special properties, some novel deceptive jamming methods against SAR using FDA have been developed. The FDA-based jamming against SAR was first studied in [24], which indicated that the direct repeater jamming without modulation using FDA can produce multiple point false targets equally spaced along the range direction. In order to improve the practical jamming performance, such as generating 2-D point false targets or multiple deceptive scenes, FDA-based deception jamming methods with convolution modulation against SAR have been proposed [25,26,27,28,29,30], while the FDA-based scattered wave deceptive jamming approach has been studied in [31,32]. Moreover, considering the fact that micro-motion modulation can form smeared ghost targets or grey line in azimuth direction [33,34,35,36], the FDA-based deception jamming methods with micro-motion modulation against SAR and SAR-GMTI were proposed in [37] and [38], respectively. However, the barrage jamming against SAR using FDA has not been considered so far.

Inspired by the unique property of FDA-based jamming and micro-motion modulation, an efficient 2-D coherent barrage jamming method against SAR is proposed in this paper. In this method, the stepped frequency offset of FDA is introduced to produce multiple barrage patches along the range direction, and the size and the distribution of the barrage patches are analyzed. Thus, combined with micro-motion modulation, the proposed technique can cause the SAR image to contain 2-D barrage jamming such as a 2-D rectangular barrage patch array, multiple range-oriented barrage stripes, or multiple azimuth-oriented barrage stripes. Thus, the target can be effectively shielded by the formed barrage jamming. Numerical results show that the distribution of the 2-D barrage jamming can be flexibly specified by properly setting the FDA parameters and the micro-motion modulation parameters properly. Moreover, the proposed jamming method will greatly benefit from the pulse compression gain since the jamming signal is coherent with the radar signal.

The remainder of the paper is organized as follows. In Section 2, the fundamental of direct repeater jamming using the FDA jammer is presented, and the influences of different change modes of frequency offset on the jamming effect are discussed. In Section 3, our barrage jamming method based on FDA and micro-motion modulation is proposed. The simulation results are given in Section 4, followed by the conclusion in Section 5.

## 2. Fundamental of Direct Repeater Jamming Using FDA Jammer

In this section, the fundamentals of the direct repeater jamming using an FDA jammer is presented, where the direct repeater means that the jamming signal is a copy of the intercepted radar signal without any modulation. More specially, the influences of different types of frequency offset on the jamming effect are examined.

### 2.1. Fixed Frequency Offset

As shown in Figure 1, the antenna of the FDA jammer is a uniform linear array containing M elements with spacing d. Taking the $\left\lceil \frac{M}{2}\right\rceil \mathrm{th}$ (⌈⋅⌉ is the round down operator) element as the reference one, the carrier frequency of the waveform radiated from each element of the FDA jammer can be represented as [20]
(1)$$f_{m}=f_{c}+\Delta f_{m}=f_{c}+\left(m-\left\lceil \frac{M}{2}\right\rceil \right)\Delta f,m=1,2,\cdots ,M$$
where $f_{c}$ is the carrier frequency of the reference element, $\Delta f_{m}$ is the frequency offset between the mth element and the reference one. Generally speaking, $\Delta f_{m}$ is a fixed linear frequency offset, i.e., the frequency increment Δf is a fixed value.

The transmit signal of the SAR can be expressed as [1]
(2)$$s\left(\tau ,\eta \right)=rect\left(\frac{\tau }{T_{p}}\right)\exp\left(j2\pi f_{c}t+j\pi k_{r}\tau^{2}\right)$$
where τ is the fast time, $T_{p}$ is the pulse width, $k_{r}$ is the chirp rate, $\eta =kT_{a}$ is the slow time, k is an integer, $T_{a}$ is the pulse repetition interval (PRI), t=τ+η is the absolute time. The intercepted signal of the jammer can be expressed as
(3)$$J_{0}\left(\tau ,\eta \right)=rect\left(\frac{\tau -R_{J}\left(\eta \right)/c}{T_{p}}\right)\exp\left(-j2\pi f_{c}\frac{R_{J}\left(\eta \right)}{c}+j\pi k_{r}{\left(\tau -\frac{R_{J}\left(\eta \right)}{c}\right)}^{2}\right)$$
where $R_{J}\left(\eta \right)=\sqrt{R_{0}{}^{2}+{\left(x_{J}-v_{a}\eta \right)}^{2}}\approx R_{0}+\frac{{\left(x_{J}-v_{a}\eta \right)}^{2}}{2R_{0}}$, $R_{0}$ is the minimum slant distance between the SAR and the FDA jammer, $v_{a}$ is the speed of the SAR, $x_{J}$ is the azimuthal coordinate of the FDA jammer, and c is the speed of light.

Ignoring the influence of the system delay and the difference of slant distance caused by element spacing, the expression of the received jamming signal transmitted by the FDA jammer is
(4)$$\begin{array}{cc} J_{1}\left(\tau ,\eta \right)= & \sum_{m=1}^{M}rect\left(\frac{\tau -2R_{J}\left(\eta \right)/c}{T_{p}}\right)\exp\left(j2\pi f_{m}\left(t-\frac{2R_{J}\left(\eta \right)}{c}\right)\right)\\ & \;\;\;\exp\left(j\pi k_{r}{\left(\tau -\frac{2R_{J}\left(\eta \right)}{c}\right)}^{2}\right) \end{array}$$

The baseband echo of the jamming signal can then be expressed as
(5)$$\begin{array}{cc} J_{2}\left(\tau ,\eta \right)= & \sum_{m=1}^{M}rect\left(\frac{\tau -2R_{J}\left(\eta \right)/c}{T_{p}}\right)\exp\left(-j\frac{4\pi R_{J}\left(\eta \right)}{\lambda }\right)\\ & \;\;\;\exp\left(j\pi k_{r}{\left(\tau -\frac{2R_{J}\left(\eta \right)}{c}\right)}^{2}+j2\pi \Delta f_{m}\left(\tau -\frac{2R_{J}\left(\eta \right)}{c}\right)\right) \end{array}$$
where $\lambda =c/f_{c}$ is the wavelength.

After range compression and range cell migration correction, the jamming signal can be described as
(6)$$J_{rc}\left(\tau ,\eta \right)\approx \sum_{m=1}^{M}\left(1-\frac{\Delta f_{m}}{B_{r}}\right)\sin c\left(B_{r}\left(\tau -\frac{2R_{0}}{c}+\frac{\Delta f_{m}}{k_{r}}\right)\right)\exp\left(-j\frac{4\pi R_{0}}{\lambda }-j\pi k_{a}{\left(\eta -\frac{x_{J}}{v_{a}}\right)}^{2}\right)$$
where $B_{r}=k_{r}T_{p}$ is the bandwidth in range domain, and the $k_{a}=\frac{2v_{a}^{2}}{\lambda R_{0}}$ represents the azimuth frequency modulation rate. After azimuth compression, we have
(7)$$J_{rd}\left(\tau ,\eta \right)=\sum_{m=1}^{M}\left(1-\frac{\Delta f_{m}}{B_{r}}\right)\sin c\left(B_{r}\left(\tau -\frac{2R_{0}}{c}+\frac{\Delta f_{m}}{k_{r}}\right)\right)\sin c\left(B_{a}\left(\eta -\frac{x_{J}}{v_{a}}\right)\right)\exp\left(-j\frac{4\pi R_{0}}{\lambda }\right)$$
where $B_{a}=\frac{2v_{a}}{L_{a}}$ is the Doppler bandwidth, $L_{a}$ is the antenna aperture of the SAR.

By observing (4) and (7), we can draw the following conclusions: The direct repeater jamming signal using the FDA jammer with fixed frequency offset is equivalent to the summation of multiple shift-frequency jamming signals with different shift-frequency values, which can generate multiple point false targets in the same azimuth location as the jammer and symmetrically and equally spaced along the range direction. The number of point false targets is determined by the number of FDA elements, and the range interval dR of adjacent false targets is $\frac{c\Delta f}{2k_{r}}$, which is proportional to the frequency increment Δf. The amplitude of each false target is proportional to the frequency increment.

Apparently, the repeater jamming using the FDA jammer with fixed linear frequency offset is too regular to be recognized. In order to increase the flexibility of repeater jamming via the FDA jammer, the FDA parameters, especially the frequency offset of each element, should be reasonably designed. Here, a random deformation of the fixed frequency offset is presented as follows
(8)$$\{\begin{array}{ll} \Delta f_{m}=\left(m-\left\lceil \frac{M}{2}\right\rceil -rand\left(0,1\right)\right)\Delta f, & m>\left\lceil \frac{M}{2}\right\rceil \\ \Delta f_{m}=\left(m-\left\lceil \frac{M}{2}\right\rceil +rand\left(0,1\right)\right)\Delta f, & m<\left\lceil \frac{M}{2}\right\rceil \\ \Delta f_{m}=0, & m=\left\lceil \frac{M}{2}\right\rceil \end{array}$$
where rand(0,1) represents a random number between 0 and 1. The image result of the repeater jamming with fixed random frequency offset is similar to (7). However, due to the presence of the random term rand(0,1), the multiple point false targets are no longer evenly spaced along the range direction.

### 2.2. Stepped Frequency Offset

In the subsection, the direct repeater jamming using the FDA jammer with stepped frequency offset is introduced.

The stepped frequency offset of FDA is defined as
(9)$$\Delta f_{m}\left(\eta \right)=\Delta f_{m0}+\partial f_{m}\frac{\left(\eta +\frac{T_{L}}{2}\right)}{T_{a}},-\frac{T_{L}}{2}\leq \eta \leq \frac{T_{L}}{2}$$
where $T_{L}$ is the synthetic aperture time, $\Delta f_{m0}$ is the initial frequency offset of the mth element, which can be a fixed linear frequency offset in the form of $\Delta f_{m}$ in (1), or a fixed random frequency offset in the form of (8), $\partial f_{m}$ is the frequency rate of the mth element, which can be different, $\partial F=\left[\partial f_{1},\partial f_{2}\cdots ,\partial f_{M}\right]$ is the frequency rate vector. The frequency offset of each element increases linearly with the slow time.

After range compression and range cell migration correction, the direct repeater jamming with stepped frequency offset can be described as
(10)$$\begin{array}{cc} J_{rc}\left(\tau ,\eta \right)= & \sum_{m=1}^{M}\left(1-\frac{\Delta f_{m}\left(\eta \right)}{B_{r}}\right)\sin c\left(B_{r}\left(\tau -\frac{2R_{0}}{c}+\frac{\Delta f_{m}\left(\eta \right)}{k_{r}}\right)\right)\cdot \\ & \;\;\;\exp\left(-j\frac{4\pi R_{J}\left(\eta \right)}{\lambda }-j\pi \frac{\Delta f_{m}^{2}\left(\eta \right)}{k_{r}}\right) \end{array}$$

It can be seen from (10) that, for the repeater jamming with stepped frequency offset, the range position of the jamming signal after range compression varies with the slow time. The range span $\Delta R_{m}$ of the jamming signal of the mth element is proportional to the variation of $\Delta f_{m}\left(\eta \right)$, which can be calculated as
(11)$$\Delta R_{m}=\frac{c\partial f_{m}T_{L}}{2T_{a}k_{r}}$$

Since the range resolution $\rho_{r}$ is $\frac{c}{2B_{r}}$, the jamming signal of the mth element spans $\partial f_{m}\frac{T_{L}}{T_{a}}T_{p}$ range cells, which means that the dwell time $T_{l_{m}}$ of the jamming signal at each range cell is $\frac{T_{a}}{\partial f_{m}T_{p}}$. According to the SAR imaging principle, if $\Delta R_{m}$ is greater than $\frac{c}{2B_{r}}$, the range displacement will lead to the image defocus in the azimuth direction. To derive the analytical expression of the image result, the jamming signal of the mth element is rewritten as
(12)$$\begin{array}{cc} J_{m}\left(\tau ,\eta \right)=\sum_{n=1}^{\partial f_{m}\frac{T_{L}}{T_{a}}T_{p}}J_{mn}\left(\tau ,\eta \right)= & \sum_{n=1}^{\partial f_{m}\frac{T_{L}}{T_{a}}T_{p}}\left(1-\frac{\Delta f_{m}\left(\eta \right)}{B_{r}}\right)\sin c\left(B_{r}\left(\tau -\frac{2R_{0}}{c}+\frac{\Delta f_{m}\left(\eta \right)}{k_{r}}\right)\right)rect\left(\frac{\eta -\Delta t_{n}}{T_{l_{m}}}\right)\cdot \\ & \;\;\;\exp\left(-j\frac{4\pi R_{0}}{\lambda }-j\pi k_{a}{\left(\eta -\frac{x_{J}}{v_{a}}\right)}^{2}\right) \end{array}$$
where $\Delta t_{n}=nT_{l_{m}}$ represents the reside moment of the jamming signal in the nth range cell.

After azimuth compression, the image result of the jamming signal of the mth element is
(13)$$\begin{array}{cc} J_{m}\left(\tau ,\eta \right)=\sum_{n=1}^{\partial f_{m}\frac{T_{L}}{T_{a}}T_{p}}J_{mn}\left(\tau ,\eta \right)= & \frac{T_{l_{m}}}{T_{a}}\left(1-\frac{\Delta f_{m}\left(\eta \right)}{B_{r}}\right)\sin c\left(B_{r}\left(\tau -\frac{2R_{0}}{c}+\frac{\Delta f_{m}\left(\eta \right)}{k_{r}}\right)\right)\cdot \\ & \sin c\left(k_{a}T_{l_{m}}\left(\eta -\frac{x_{J}}{v_{a}}\right)\right)\exp\left(-j\frac{4\pi R_{0}}{\lambda }\right) \end{array}$$

Furthermore, the image result of the direct repeater jamming with steeped frequency offset can be obtained as
(14)$$\begin{array}{cc} J_{rd}\left(\tau ,\eta \right)=\sum_{m=1}^{M}J_{m}\left(\tau ,\eta \right)= & \sum_{m=1}^{M}\frac{T_{l_{m}}}{T_{a}}\left(1-\frac{\Delta f_{m}\left(\eta \right)}{B_{r}}\right)\sin c\left(B_{r}\left(\tau -\frac{2R_{0}}{c}+\frac{\Delta f_{m}\left(\eta \right)}{k_{r}}\right)\right)\cdot \\ & \;\;\;\sin c\left(k_{a}T_{l_{m}}\left(\eta -\frac{x_{J}}{v_{a}}\right)\right)\exp\left(-j\frac{4\pi R_{0}}{\lambda }\right) \end{array}$$

By observing (13) and (14), we can draw the conclusion that the direct repeater jamming using the FDA jammer with stepped frequency offset can create up to M rectangular barrage patches distributed along the range direction. The azimuth coordinate of the center of all patches is the same as the azimuth position of the jammer. The start range position and the end range position of the patch corresponding to the mth element can be expressed as follows
(15)$$R_{mS}=R_{0}-\frac{\Delta f_{m0}c}{2k_{r}}$$
(16)$$R_{mE}=R_{0}-\frac{\Delta f_{m0}c}{2k_{r}}-\frac{\partial f_{m}T_{L}c}{2k_{r}T_{a}}$$

The range span and azimuth span of the patch can be expressed as follows
(17)$$\begin{array}{l} \Delta R_{m}=\frac{\partial f_{m}T_{L}c}{2k_{r}T_{a}}=\partial f_{m}\frac{T_{L}}{T_{a}}T_{p}\rho_{r}\\ \Delta x_{m}=\frac{v_{a}}{k_{a}T_{l_{m}}}=\partial f_{m}\frac{T_{L}}{T_{a}}T_{p}\rho_{a} \end{array}$$
where $\rho_{a}=\frac{v_{a}}{B_{a}}$ is the azimuth resolution. Obviously, the size of the patch is proportional to the square of the frequency rate $\partial f_{m}$.We can also obtain
(18)$$\frac{\Delta R_{m}}{\Delta x_{m}}=\frac{\rho_{r}}{\rho_{a}}$$

Equation (18) means that the ratio of range span to azimuth span of the patch is always equal to the range-azimuth resolution ratio of the SAR.

## 3. The Proposed Barrage Jamming Method against SAR

Section 2 indicates that the direct repeater jamming using the FDA jammer can generate multiple point false targets or rectangular barrage patches, but only along the range direction, so improvement needs to be made so that the jamming signal of the FDA jammer is capable of generating point false targets or rectangular barrage patches that expend not only along the range direction but also along the azimuth direction.

As is known, the micro-motion modulation jamming can produce multiple point false targets or barrage lines along the azimuth direction. Thus, by combining FDA jammer and micro-motion modulation, an efficient 2-D barrage jamming is proposed in this section.

### 3.1. Imaging Model of the Proposed Method

The proposed barrage jamming signal can be expressed as
(19)$$\begin{array}{cc} J_{1}\left(\tau ,\eta \right)= & \sum_{m=1}^{M}rect\left(\frac{\tau -2R_{J}\left(\eta \right)/c}{T_{p}}\right)\exp\left(-j\frac{4\pi R_{J}\left(\eta \right)}{\lambda }\right)\cdot \\ & \;\;\;\exp\left(j\pi k_{r}{\left(\tau -\frac{2R_{J}\left(\eta \right)}{c}\right)}^{2}+j2\pi \Delta f_{m}\left(\eta \right)\left(\tau -\frac{2R_{J}\left(\eta \right)}{c}\right)\right)\cdot \\ & \;\;\;\exp\left(-jA\sin\left(\omega \eta \right)\right) \end{array}$$
where exp(−jAsin(ωη)) is the micro-motion modulation signal [35,36,37], A and ω is the modulation amplitude and modulation angular velocity. According to the Bessel expansion given as [35]
(20)$$\exp\left(-jx\sin\theta \right)=\sum_{n=-\infty }^{+\infty }J_{n}\left(x\right)\exp\left(jn\theta \right)$$
where $J_{n}\left(\cdot \right)$ represent the n−order Bessel function of the first kind, (19) can be rewritten as
(21)$$\begin{array}{cc} {J^{\prime }}_{1}\left(\tau ,\eta \right)= & \sum_{m=1}^{M}rect\left(\frac{\tau -2R_{J}\left(\eta \right)/c}{T_{p}}\right)\exp\left(-j\frac{4\pi f_{c}R_{J}\left(\eta \right)}{c}\right)\cdot \\ & \;\;\;\exp\left(j\pi k_{r}{\left(\tau -\frac{2R_{J}\left(\eta \right)}{c}\right)}^{2}+j2\pi \Delta f_{m}\left(\eta \right)\left(\tau -\frac{2R_{J}\left(\eta \right)}{c}\right)\right)\cdot \\ & \;\;\;\sum_{n=-\infty }^{+\infty }J_{n}\left(A\right)\exp\left(-jn\omega \eta \right) \end{array}$$

For the fixed frequency offset, according to the micro-motion jamming effect [35,36,37], the image result of the proposed jamming can be derived as
(22)$$\begin{array}{cc} J_{rd}\left(\tau ,\eta \right)= & \sum_{m=1}^{M}\left(1-\frac{\Delta f_{m}}{B_{r}}\right)\sin c\left(B_{r}\left(\tau -\frac{2R_{0}}{c}+\frac{\Delta f_{m}}{k_{r}}\right)\right)\cdot \\ & \;\;\;\sum_{n=-\infty }^{+\infty }J_{n}\left(A\right)\left(1-\left|\frac{n\omega }{2\pi B_{d}}\right|\right)\sin c\left(B_{a}\left(\eta -\frac{x_{J}}{v_{a}}-\frac{n\omega }{2\pi k_{a}}\right)\right)\exp\left(-j\frac{4\pi R_{0}}{\lambda }-j\pi {\left(\frac{n\omega }{2\pi k_{a}}\right)}^{2}\right) \end{array}$$

By observing (22), we can draw the following conclusion: for the fixed frequency offset, the repeater jamming using the FDA jammer based on micro-motion modulation can generate 2-D multiple point false targets evenly distributed along the azimuth direction at M range positions, which are $R_{0}+\frac{\Delta f_{m}c}{2k_{r}}\left(m=1,2\cdots ,M\right)$. The azimuth interval of adjacent false targets, and the total number and cover length of multiple false targets along the azimuth can be expressed as
(23)$$dx=\frac{\omega v_{a}}{2\pi k_{a}}$$
(24)$$Num_{J}=2\left|A\right|+1$$
(25)$$L_{J}=2\left|A\right|dx$$

When dx is greater than $\rho_{a}$, the jamming effect is presents as 2-D multiple point false targets, otherwise, it is presented as multiple azimuth-oriented barrage lines.

For the stepped frequency offset, the image results can be expressed as
(26)$$\begin{array}{cc} J_{rd}\left(\tau ,\eta \right)= & \sum_{m=1}^{M}\frac{T_{l_{m}}}{T_{a}}\left(1-\frac{\Delta f_{m}\left(\eta \right)}{B_{r}}\right)\sin c\left(B_{r}\left(\tau -\frac{2R_{0}}{c}+\frac{\Delta f_{m}\left(\eta \right)}{k_{r}}\right)\right)\cdot \\ & \;\;\;\sum_{n=-\infty }^{+\infty }J_{n}\left(A\right)\left(1-\left|\frac{n\omega }{2\pi B_{d}}\right|\right)\sin c\left(k_{a}T_{l_{m}}\left(\eta -\frac{x_{J}}{v_{a}}-\frac{n\omega }{2\pi k_{a}}\right)\right)\exp\left(-j\frac{4\pi R_{0}}{\lambda }-j\pi {\left(\frac{n\omega }{2\pi k_{a}}\right)}^{2}\right) \end{array}$$

By observing (26), it can be seen that for the stepped frequency offset, the repeater jamming using FDA jammer based on micro-motion modulation can generate 2-D multiple rectangular barrage patches evenly distributed along the azimuth direction at M range positions, which are $R_{0}-\frac{\Delta f_{m0}c}{2k_{r}}\left(m=1,2\cdots ,M\right)$, and the azimuth interval dx of adjacent patches is $\frac{\omega v_{a}}{2\pi k_{a}}$. The number of patches along the azimuth is 2|A|+1, and the size of each patch is given by (17).

### 3.2. Jamming Results Analysis

As shown in Section 3.1, when the frequency offset is stepped, the proposed jamming method is able to form a 2-D barrage jamming consisting of multiple rectangular patches in the SAR image. Therefore, in this subsection, the analysis will focus on the jamming effect of the proposed method when the stepped frequency offset with fixed random initial frequency offset and unique frequency rate is used, and more details will be illustrated in the simulation.

As shown in the previous sections, the start range position, range span, azimuth span, azimuth interval, number and azimuth coverage of the patches corresponding to the mth element are given as follows
(27)$$R_{mS}=R_{0}-\frac{\Delta f_{m0}c}{2k_{r}}$$
(28)$$\Delta R_{m}=\frac{\partial f_{m}T_{L}c}{2k_{r}T_{a}}$$
(29)$$\Delta x_{m}=\frac{v_{a}}{k_{a}T_{l_{m}}}=\frac{\Delta R_{m}\rho_{a}}{\rho_{r}}$$
(30)$$dx=\frac{\omega v_{a}}{2\pi k_{a}}=\frac{\omega \lambda R_{0}}{4\pi v_{a}}$$
(31)$$Num_{J}=2\left|A\right|+1$$
(32)$$\Delta x_{Jc}=2\left|A\right|dx+\Delta x_{m}$$

Furthermore, the range interval of the patches corresponding to the adjacent elements can be calculated as
(33)$$dR=\frac{\Delta fc}{2k_{r}}$$

From (28), (29), (30) and (33), we can draw the following conclusion. If both the range interval dR and the azimuth interval dx of adjacent patches are greater than the range span $\Delta R_{m}$ and the azimuth span $\Delta x_{m}$ of the patch, respectively, the jamming effect is presented as 2-D rectangle barrage patch array composed of patches of the same size. If dR is less than $\Delta R_{m}$ and dx is greater than $\Delta x_{m}$ of the patch, the jamming effect appears as multiple range barrage strips of the same length and width. If dR is greater than $\Delta R_{m}$ and dx is less than $\Delta x_{m}$, the jamming effect appears as multiple azimuth-oriented barrage strips of the same length and width. If both dR and dx are less than $\Delta R_{m}$ and $\Delta x_{m}$, the jamming effect degenerates to a whole large barrage patch.

Considering that SAR targets are mostly distributed targets, the proposed jamming method can only achieve the jamming effect if the target is covered by one or more of these patches, and therefore reasonable settings for the FDA parameters and micro-motion parameters are required. A brief analysis is given below to illustrate the setting of the jamming parameters. Assuming that the range span and azimuth span of the target are $\Delta R_{T}$ and $\Delta x_{T}$ respectively, if the target is covered by one of the patches in the range direction, the frequency rate $\partial f_{m}$ is subject to the following condition
(34)$$\partial f_{m}\geq \frac{2k_{r}T_{a}\Delta R_{T}}{T_{L}c}$$

At this point, the azimuth span $\Delta x_{m}$ of the batch can be expressed $\frac{\partial f_{m}T_{L}c}{2k_{r}T_{a}}\frac{\rho_{a}}{\rho_{r}}$. If $\Delta x_{m}$ is greater or equal to $\Delta x_{T}$, the setting of the micro-motion modulation parameters does not require much consideration. Otherwise, multiple patches need to aliased into a large enough patch to cover the target in the azimuth direction, so the micro-motion parameters must satisfy the following conditions
(35)$$dx=\frac{\omega v_{a}}{2\pi k_{a}}\leq \Delta x_{m}=\frac{\partial f_{m}^{}T_{L}c}{2k_{r}T_{a}}\frac{\rho_{a}}{\rho_{r}}$$
(36)$$\Delta x_{Jc}=2\left|A\right|dx+\Delta x_{m}\geq \Delta x_{T}$$

In turn, we can obtain that
(37)$$\omega \leq \frac{2\pi k_{a}\Delta x_{m}}{v_{a}}=\frac{\pi k_{a}\partial f_{m}^{}T_{L}c}{k_{r}T_{a}v_{a}}\frac{\rho_{a}}{\rho_{r}}$$
(38)$$\left|A\right|\geq \left\lfloor \frac{\Delta x_{T}-\Delta x_{m}}{2dx}\right\rfloor =\left\lfloor \left(\Delta x_{T}-\frac{\partial f_{m}^{}T_{L}c}{2k_{r}T_{a}}\frac{\rho_{a}}{\rho_{r}}\right)/\left(\frac{\omega v_{a}}{\pi k_{a}}\right)\right\rfloor$$
where ⌊⋅⌋ is the round up operator. Setting the jamming parameters in other cases can be done as described above and is not repeated here.

### 3.3. Analysis of the Influence of Reconnaissance Errors

In this subsection, the influence of reconnaissance errors on the jamming effect is analyzed. As shown in Section 3.2, the jamming effect is mainly related to the chirp rate $k_{r}$, the PRI $T_{a}$, the synthetic aperture time $T_{L}$, the wavelength λ and the speed $v_{a}$ of the SAR. Suppose the reconnaissance values of the above parameters are $k_{r}+\Delta k_{r}$, $T_{a}+\Delta T_{a}$, $T_{L}+\Delta T_{L}$, λ+Δλ and $v_{a}+\Delta v_{a}$, respectively, and the relative reconnaissance errors are $\varepsilon_{k_{r}}=\Delta k_{r}/k_{r}$, $\varepsilon_{T_{a}}=\Delta T_{a}/T_{a}$, $\varepsilon_{T_{L}}=\Delta T_{L}/T_{L}$, $\varepsilon_{\lambda }=\Delta \lambda /\lambda$ and $\varepsilon_{v_{a}}=\Delta v_{a}/v_{a}$, respectively. The start range position, range span, and azimuth interval of the patches corresponding to the mth element can then be expressed as
(39)$${R^{*}}_{mS}=R_{0}-\frac{\left(1+\varepsilon_{k_{r}}\right)\Delta f_{m0}c}{2k_{r}}$$
(40)$${\Delta R_{m}}^{*}=\frac{\left(1+\varepsilon_{T_{a}}\right)\left(1+\varepsilon_{k_{r}}\right)}{\left(1+\varepsilon_{T_{L}}\right)}\frac{\partial f_{m}T_{L}c}{2k_{r}T_{a}}$$
(41)$${dx}^{*}=\frac{\left(1+\varepsilon_{v_{a}}\right)}{\left(1+\varepsilon_{\lambda }\right)}\frac{\omega \lambda R_{0}}{4\pi v_{a}}$$

From (39), (40) and (41), it can be seen that the range interval between the patches corresponding to the mth element and the patches corresponding to the reference element is proportional to $\varepsilon_{k_{r}}$, the range span of the patch is proportional to $\varepsilon_{k_{r}}$ and $\varepsilon_{T_{a}}$ and inversely proportional to $\varepsilon_{T_{L}}$, and the azimuth interval of the patches is proportional to $\varepsilon_{v_{a}}$ and inversely proportional to $\varepsilon_{\lambda }$.

## 4. Results

In this section, the jamming effect of the direct repeater jamming using the FDA jammer and the proposed barrage jamming are validated by extensive simulation results.

### 4.1. Validation of Direct Repeater Jamming Using the FDA Jammer

According to the theoretical analysis, the direct repeater jamming using the FDA jammer can produce multiple point false targets or multiple rectangular barrage patches along the range direction, which will be present in the subsection. The parameters of the SAR system are listed in Table 1. The jammer is fixed in the center of the scene.

#### 4.1.1. Multiple Point False Targets along the Range Direction

Figure 2 presents the jamming results of direct repeater jamming using an FDA jammer with fixed linear frequency offset, where the left subgraphs and the right subgraphs are the 2-D image and the range profile of the point false targets, respectively. As shown in Figure 2, the direct repeater jamming can produce multiple point false targets that are symmetrical and equally spaced along the range direction. The numbers of false targets in each subgraph are 7, 7 and 6, which is equal to the number of FDA elements. The range intervals of adjacent false targets in each subgraph are 75 m, 300 m and 300 m, which is proportional to the frequency increment. The jamming results are consistent with the theoretical analysis in Section 2.1. The amplitude of the false targets is inversely proportional to the frequency increment, and decreases from the center of the symmetry to both sides.

Moreover, the jamming result of the direct repeater jamming using the FDA jammer with fixed random frequency offset is shown in Figure 3. Obviously, the false targets are not equally distributed along the range direction, and there is no symmetry in the variation of the false target amplitude, which increases the diversity of the jamming effect to some extent.

#### 4.1.2. Multiple Rectangular Barrage Patches along the Range Direction

In the subsection, the jamming effect of the repeater jamming using an FDA jammer with stepped frequency offset which can produce multiple rectangular barrage patches along the range direction is validated. The number of FDA elements is set as seven.

Figure 4 shows the jamming results of the direct repeater jamming with stepped frequency offset at various range-azimuth resolution ratios, where the initial frequency offset of each element is fixed linear and the frequency rate of all elements is the same. The pulse widths of the SAR in Figure 4a,b are set as 15 us and 30 us, respectively. The other parameters are consistent with those shown in Table 1. The corresponding range-azimuth resolution ratios of Figure 4a,b are 1 m and 0.5 m, respectively. As shown in Figure 4, the range span and azimuth span of each patch can be calculated as 50 m and 50 m in Figure 4a, respectively, while the range span and azimuth span of each patch are 50 m and 100 m in Figure 4b, respectively, which is consistent with (17) and (18). The result validates the conclusion that the ratio of range span to azimuth span of the patch is always equal to the range-azimuth resolution ratio of the SAR.

Figure 5 presents the jamming results of the direct repeater jamming with stepped frequency offset and fixed linear initial frequency offset and unique frequency rate. As shown in Figure 5, the rectangular barrage patches are equally spaced along the range direction, the range span of each rectangular barrage patch is proportional to the frequency rate ∂f, and the range interval of adjacent patches is proportional to the frequency increment Δf. In particular, as shown in Figure 5d, the range interval of adjacent patches is smaller than the range span of each patch ($\left|R_{mS}-R_{m+1S}\right|=\frac{\Delta fc}{2k_{r}}=12.5\;m<$
$\Delta R_{m}=\frac{\partial f_{m}T_{L}c}{2k_{r}T_{a}}=50\;m$), all the patches are aliased into a large patch distributed along the range direction.

Figure 6 shows the influence of fixed random initial frequency offset on the jamming effect. It can be seen that the barrage patches with the same size are unevenly spaced along the range direction, and even some adjacent patches are aliased. Figure 7 shows the influence of the non-uniqueness of the frequency rates on the jamming effect, from which it can be seen that the barrage patches are not of the same size, where the range span of the patches are 25 m, 50 m, 35 m, 75 m, 25 m, 50 m and 35 m, respectively. In other words, the fixed random initial frequency offset and the non-uniqueness of the frequency rates can increase the flexibility of the jamming effect.

### 4.2. Validation of the Proposed Barrage Jamming Method

Theoretical analysis indicates that the proposed coherent barrage jamming method combined with the FDA jammer and micro-motion modulation can produce 2-D point false targets or rectangular barrage patches. In the subsection, simulated data is utilized to validate the proposed method, and the influence of frequency offset on the range direction distribution characteristic of the image results of the proposed coherent barrage jamming is consistent with that mentioned above and will not be described here. The number of FDA elements is set as 7. The parameters of the SAR system are listed in Table 1. Figure 8 shows the original SAR image with a tank-shaped target in the center of the scene. The range span and azimuth span of the area of the target are approximately 10 m and 16 m, respectively.

#### 4.2.1. 2-D Point False Targets

Figure 9 presents the image results of the proposed jamming with fixed frequency offset, where the left subgraphs and the right subgraphs are the global perspective and partial enlargement, respectively. The jammer-to-signal ratio (JSR) is 10 dB. When the azimuth interval of adjacent false targets is smaller than the azimuth resolution ($dx=\frac{\omega v_{a}}{2\pi k_{a}}=0.8\;m<\rho_{a}=\frac{L_{a}}{2}=1\;m$), the jamming effect is present as multiple azimuth-oriented barrage lines located at different range positions as shown in Figure 9a,b; otherwise, the jamming effect is present as a 2-D point false target array, as shown in Figure 9c, where dx is greater than $\rho_{a}$($dx=40\;m,\;\rho_{a}=1\;m$). The cover length of multiple false targets or barrage lines along the azimuth is proportional to the modulation amplitude and modulation angular velocity of the micro-motion modulation signal. What needs to be pointed out is that the proposed jamming with fixed frequency offset is not suitable for protecting surface targets such as tanks.

#### 4.2.2. 2-D Rectangular Barrage Patches

Considering that the imaging area of the target is 10 m×16 m, it can be easily be covered by a single patch. As the range span of the patch is the same as the azimuth span in the simulation, it is only necessary set the frequency offset parameters to ensure that the range span of the patch is greater than 16 m. According to (34) in Section 3.2, it can concluded that the frequency rate $\partial f_{m}$ should be greater than $\frac{32}{9}\;\mathrm{KHz}$. In addition, the patches corresponding to the reference element are shifted back 10 m in the range direction by a tiny time delay $\frac{1}{15}\;\mu s$ to cover the target. The JSR is set as 18 dB

As shown in Figure 10, the proposed jamming with stepped frequency offset had a fixed linear initial frequency offset and a unique frequency rate, and can produce 2-D rectangular barrage patches of the same size evenly distributed along both the range and the azimuth directions, where the distribution characteristics of the barrage patches along the azimuth direction and the range direction are decided by the micro-motion modulation parameters and the FDA parameters, respectively. As shown in Figure 10a, the range interval dR and the azimuth interval dx of adjacent patches are 100 m and 50 m, respectively, the range span $\Delta R_{m}$ and the azimuth span $\Delta x_{m}$ of each patch are 25 m and 25 m, respectively, which means both dR and dx are greater than $\Delta R_{m}$ and $\Delta x_{m}$, respectively, the jamming effect is presented as a 2-D rectangular barrage patch array composed of patches of the same size. As shown in Figure 10b, dR, dx, $\Delta R_{m}$ and $\Delta x_{m}$ are 22.5 m, 50 m, 25 m and 25 m, respectively, i.e., dR is less than $\Delta R_{m}$ and dx is greater than $\Delta x_{m}$, the jamming effect is presented as multiple range barrage stripes of equal length and width. As shown in Figure 10c, dR, dx, $\Delta R_{m}$ and $\Delta x_{m}$ are 75 m, 10 m, 25 m and 25 m, respectively, i.e., dR is greater than $\Delta R_{m}$ and dx is less than $\Delta x_{m}$, the jamming effect is presented as multiple azimuth-oriented barrage stripes with same length and width. As shown in Figure 10d, dR, dx, $\Delta R_{m}$ and $\Delta x_{m}$ are 22.5 m, 10 m, 25 m and 25 m, respectively, i.e., dR is less than $\Delta R_{m}$ and dx is less than $\Delta x_{m}$, the jamming effect degenerates into a single large barrage patch.

Figure 11 shows the influence of the non-uniqueness of the frequency rates on the jamming effect of the proposed jamming. It can be seen that the size of the patches is different at different range positions. As a result, the jamming effect is presented as a 2-D rectangular barrage patch array composed of patches of different sizes, or multiple range barrage stripes with uneven widths, or multiple azimuth-oriented barrage stripes with different widths, or a combination of multiple barrage stripes and multiple barrage arrays.

Figure 12 shows the jamming results under different parameter reconnaissance errors. Compared with the jamming result when all the parameters are accurate as shown in Figure 12a, the relative error $\varepsilon_{k_{r}}$ will cause the range interval of adjacent patches to increase to 150 m and the range span of the patch to increase to 37.5 m, as shown in Figure 12b; $\varepsilon_{T_{a}}$ will cause the range span of the patch to increase to 37.5 m, as shown in Figure 12c; $\varepsilon_{T_{L}}$ will cause the range span of the patch to decrease to 16.7 m, as shown in Figure 12d; $\varepsilon_{\lambda }$ will cause the azimuth interval of adjacent patches to decrease to 33.3 m; $\varepsilon_{v_{a}}$ will cause the azimuth interval of adjacent patches to increase to 75 m. The simulation results are consistent with the theoretical analysis. Thus, the proposed jamming method can be well adapted to the parameter reconnaissance errors.

To sum up, in contrast to the direct repeater jamming with fixed frequency offset which can only generate point false targets distributed along the range direction, the proposed method creates multiple barrage patches in the range direction by introducing the stepped frequency offset, and extends the coverage of the barrage patches to the azimuth direction by combining micro-motion modulation, resulting in a 2-D barrage jamming with variable effects. Moreover, by setting reasonable jamming parameters, the proposed jamming with stepped frequency offset can not only suppress the surface target directly, but also form cover patches around a real target to further improve the confusion of jamming.

## 5. Conclusions

In this paper, a 2-D coherent barrage jamming method against SAR is proposed. By controlling the change mode of the frequency offset of the FDA jammer and utilizing micro-motion modulation in slow time, the jamming signal can form a flexible and controllable barrage jamming effect in the SAR image. If the frequency offset of FDA is fixed, the SAR image can be covered by a 2-D point false target array or multiple azimuth-oriented barrage lines; if the frequency offset of FDA is stepped, the real scene can be covered by a 2-D rectangular barrage patch array, multiple range-oriented barrage stripes or multiple azimuth-oriented barrage stripes. The numerical simulation results demonstrate the effectiveness of the proposed method. Furthermore, due to the existence of azimuth barrage patches, it may also be effective against SAR-GMTI. The application of the proposed jamming method against SAR-GMTI will be another important direction for future work.

## Figures and Tables

**Figure 1 sensors-23-02449-f001:**
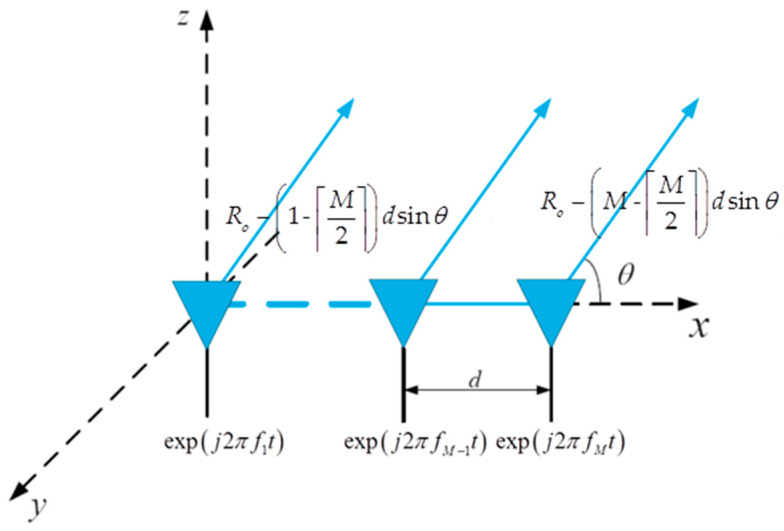
FDA geometry model.

**Figure 2 sensors-23-02449-f002:**
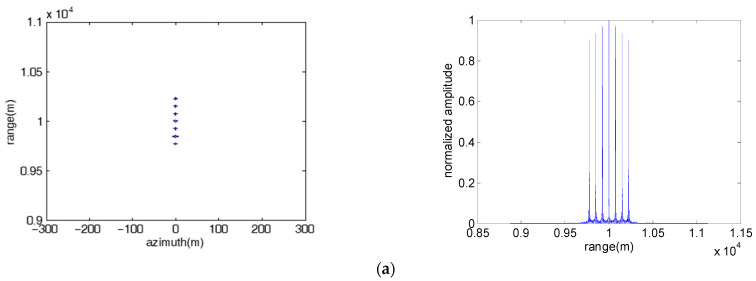
Image results of the direct repeater jamming with fixed linear frequency offset. (**a**) M=7, Δf=5 MHz; (**b**) M=7, Δf=20 MHz; (**c**) M=6, Δf=20 MHz.

**Figure 3 sensors-23-02449-f003:**
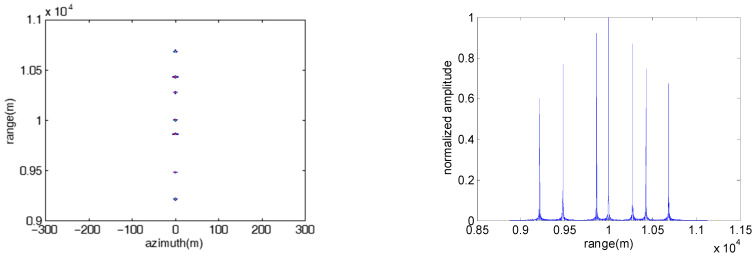
Image result of the direct repeater jamming with fixed random frequency offset. M=7, Δf=20 MHz.

**Figure 4 sensors-23-02449-f004:**
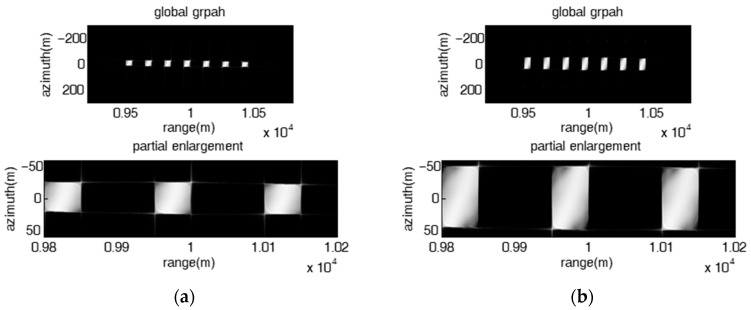
Image results of the direct repeater jamming with stepped frequency offset at various range-azimuth resolution ratios. Δf=10 MHz, $\partial f=\frac{100}{9}\;\mathrm{KHz}$. (**a**) $\frac{\rho_{r}}{\rho_{a}}=1$; (**b**) $\frac{\rho_{r}}{\rho_{a}}=0.5$.

**Figure 5 sensors-23-02449-f005:**
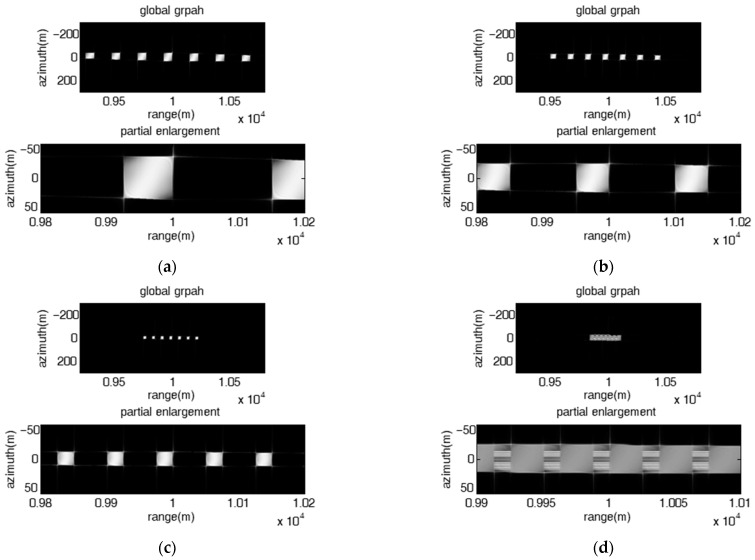
Image results of the direct repeater jamming with stepped frequency offset had a fixed linear initial frequency offset and a unique frequency rate. (**a**) Δf=15 MHz, $\partial f=\frac{50}{3}\;\mathrm{KHz}$; (**b**) Δf=10 MHz, $\partial f=\frac{100}{9}\;\mathrm{KHz}$; (**c**) Δf=5 MHz, $\partial f=\frac{50}{9}\;\mathrm{KHz}$; (**d**) Δf=2.5 MHz, $\partial f=\frac{100}{9}\;\mathrm{KHz}$.

**Figure 6 sensors-23-02449-f006:**
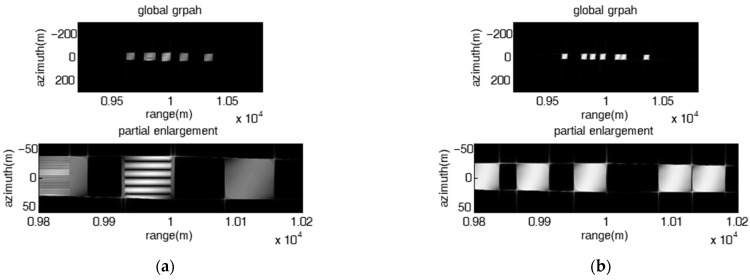
Image results of the direct repeater jamming with stepped frequency offset had a fixed random initial frequency offset and a unique frequency rate. (**a**) Δf=10 MHz, $\partial f=\frac{50}{3}\;\mathrm{KHz}$; (**b**) Δf=10 MHz, $\partial f=\frac{100}{9}\;\mathrm{KHz}$.

**Figure 7 sensors-23-02449-f007:**
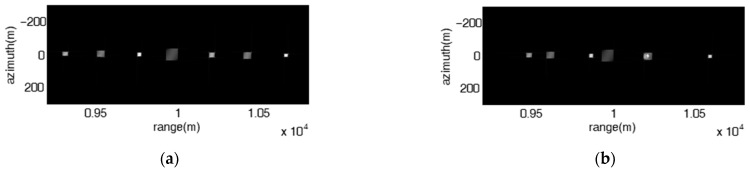
Image results of the direct repeater jamming with stepped frequency offset had various frequency rates, Δf=15 MHz, $\partial F=\left[\frac{50}{9},\frac{100}{9},\frac{70}{9},\frac{50}{3},\frac{50}{9},\frac{100}{9},\frac{70}{9}\right]\;\mathrm{KHz}$. (**a**) fixed linear initial frequency offset; (**b**) fixed random initial frequency offset.

**Figure 8 sensors-23-02449-f008:**
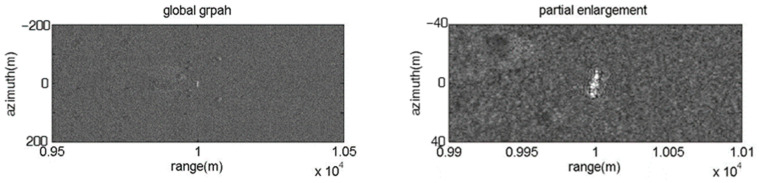
Image result of real scene.

**Figure 9 sensors-23-02449-f009:**
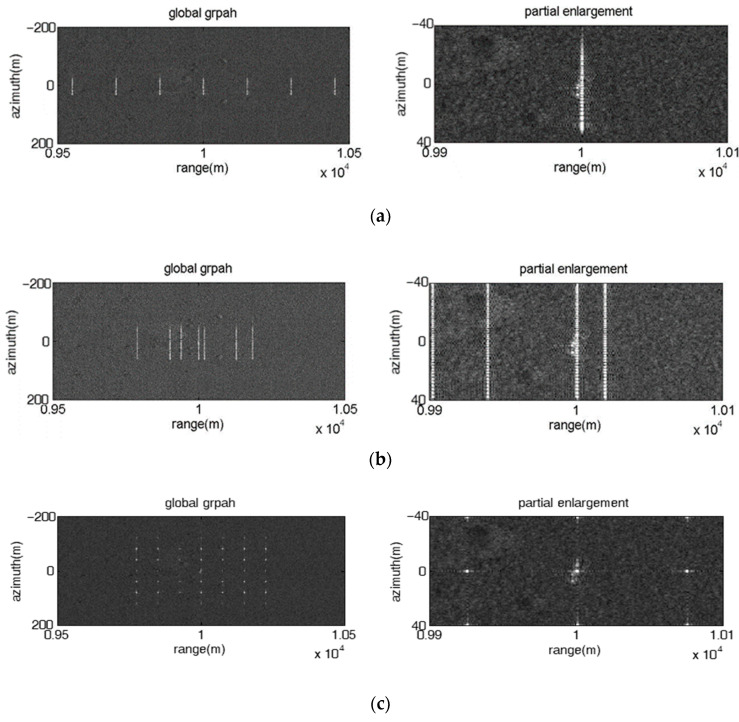
Image results of the proposed jamming with fixed frequency offset. (**a**) fixed linear frequency offset, $\Delta f=10\;\mathrm{MHz},\;A=37.5,\;\omega =\frac{32\pi }{15}$; (**b**) fixed random frequency offset $\Delta f=5\;\mathrm{MHz},\;A=75,\;\omega =\frac{32\pi }{15}$; (**c**) fixed linear frequency offset, $\Delta f=5\;\mathrm{MHz},\;A=3,\;\omega =\frac{320\pi }{3}$.

**Figure 10 sensors-23-02449-f010:**
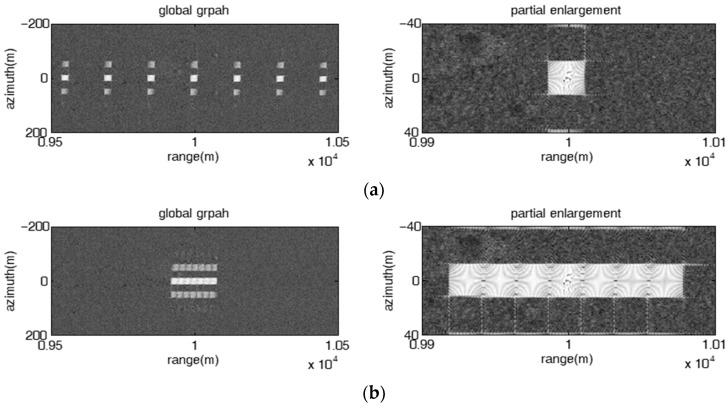
Image results of the proposed jamming with stepped frequency offset had fixed linear initial frequency offset and unique frequency rate. (**a**) $\Delta f=10\;\mathrm{MHz},\;\partial f=\frac{50}{9}\;\mathrm{KHz},\;A=1,\;\omega =\frac{400}{3}\pi$; (**b**) $\Delta f=1.5\;\mathrm{MHz},\;\partial f=\frac{50}{9}\;\mathrm{KHz},\;A=1,\;\omega =\frac{400}{3}\pi$; (**c**) $\Delta f=5\;\mathrm{MHz},\;\partial f=\frac{50}{9}\;\mathrm{KHz},\;A=9,\;\omega =\frac{80\pi }{3}$; (**d**) $\Delta f=1.5\;\mathrm{MHz},\;\partial f=\frac{50}{9}\;\mathrm{KHz},\;A=9,\;\omega =\frac{80\pi }{3}$.

**Figure 11 sensors-23-02449-f011:**
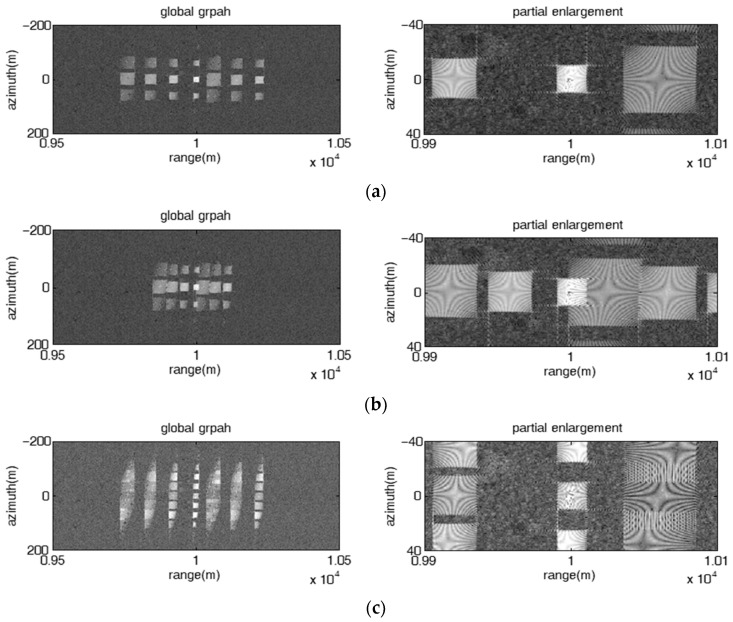
Image results of the proposed jamming with stepped frequency offset had various frequency rates. (**a**) $\Delta f=5\;\mathrm{MHz},\;\partial F=\left[\frac{60}{9},\frac{80}{9},\frac{100}{9},\frac{40}{9},\frac{60}{3},\frac{80}{9},\frac{100}{9}\right]\;\mathrm{KHz},\;A=1,\;\omega =\frac{480\pi }{3}$; (**b**) $\Delta f=2.5\;\mathrm{MHz},\;\partial F=\left[\frac{60}{9},\frac{80}{9},\frac{100}{9},\frac{40}{9},\frac{60}{3},\frac{80}{9},\frac{100}{9}\right]\;\mathrm{KHz},\;A=1,\;\omega =\frac{480\pi }{3}$; (**c**) Δf=2.5 MHz,
$\partial F=\left[\frac{60}{9},\frac{80}{9},\frac{100}{9},\frac{40}{9},\frac{60}{3},\frac{80}{9},\frac{100}{9}\right]\;\mathrm{KHz},\;A=3,\;\omega =\frac{280\pi }{3}$; (**d**) Δf=5 MHz,
$\partial F=\left[\frac{60}{9},\frac{80}{9},\frac{100}{9},\frac{40}{9},\frac{60}{3},\frac{80}{9},\frac{100}{9}\right]\;\mathrm{KHz},\;A=1,\;\omega =\frac{480\pi }{3}$.

**Figure 12 sensors-23-02449-f012:**
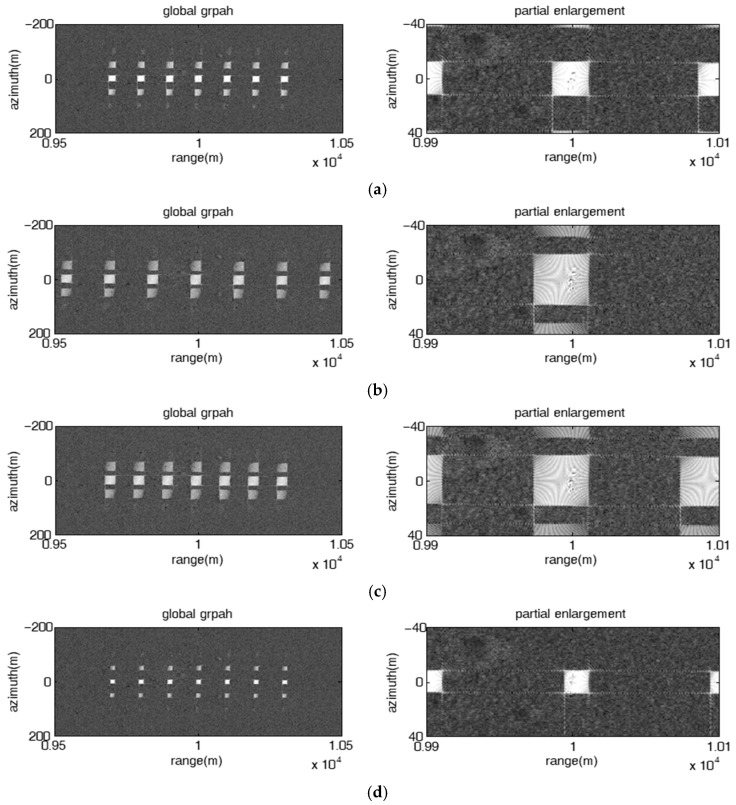
Image results of the proposed jamming with stepped frequency offset under different reconnaissance errors. $\Delta f=\frac{20}{3}\;\mathrm{MHz},\;\partial f=\frac{50}{9}\;\mathrm{KHz},\;A=1,\;\omega =\frac{480\pi }{3}$. (**a**) No reconnaissance errors; (**b**) $\varepsilon_{k_{r}}=50\%$; (**c**) $\varepsilon_{T_{a}}=50\%$; (**d**) $\varepsilon_{T_{L}}=50\%$; (**e**) $\varepsilon_{\lambda }=50\%$; (**f**) $\varepsilon_{v_{a}}=50\%$.

**Table 1 sensors-23-02449-t001:** Parameters of the SAR system.

Parameters	Value
Carrier frequency	10 G Hz
Pulse width	15 us
Chirp rate	10^13^ Hz/s
Reference slant range	10 km
Azimuth resolution	1 m
Synthetic aperture time	0.75 s
Platform velocity	200 m/s
Doppler bandwidth	200 Hz
Pulse repetition frequency	400 Hz

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
