# Peer review of "Two-Dimensional Barrage Jamming against SAR Using a Frequency Diverse Array Jammer"

_sensors, 2023, doi:10.3390/s23052449_

Round 1
Reviewer 1 Report
This manuscript proposed a barrage jamming method against SAR Using FDA jammer. The topic is interesting and relevant. The introduction provides sufficient background and include most relevant references. The proposed method has some novelty and could be considered for application in the future. However, the manuscript could not be published until the following questions are clearly clarified.
1. There are some type errors, such as, “the left subgraph and the left subgraph” in line 231 and line 318, “In the subsubsection” in line 256, “tow-dimensional” in line 399.
2. “By observing (22)” in line 211 should be “(26)”.
3. In section 2.1, the difference of slant distance caused by element spacing was ignored. However, the antenna does not always point to normal direction. In this case, the assumption is not valid.
4. There are some errors in the equations reference in section 2.2.
5. In section 4.1, the parameters of FDA jammer also need to be listed in detail.
6. It is better the use same parameters in sections 4.2.1 and 4.2.2, such as JNR.
Reviewer 2 Report
This manuscript proposes a barrage jamming method against SAR using the FDA jammer with stepped frequency offset and micro-motion modulation. Various simulation results are presented to validate the proposed method. In general, this manuscript cannot be accepted without a major revision. The comments are as follows.
1. The writing of this manuscript needs a significant improvement, there are lots of problems of grammar, spelling, symbol definition, symbol consistency, equation number, etc.
2. It is suggested to give the references for some equations to distinguish the cited/classical ones and the original ones formulated by this manuscript. Besides, all the equations should be checked to make sure they are correct. For example, Eqs. (13) and (14) maybe incorrect.
3. Some discussions of the methods/principles to set the jamming parameters should be provided. For example, if the jammer position is different from the target position, how to set the jamming parameters to effectively shield the target?
4. It is suggested to provide some results with real-measured SAR data, which should not be difficult, and some quantitative performance analysis of the proposed jamming method.
5. It is known that the parameters of the repeater jamming signal cannot be exactly the same with the SAR signal, e.g., the chirp rate, the slow time, the PRI, etc. So, it is suggested to provide some results and corresponding discussions of the jamming performance of the proposed method with parameters errors.
Reviewer 3 Report
Inspired by the unique property of FDA-based jamming and Micro-motion modulation, an efficient 2-D coherent barrage jamming method against SAR is proposed in this paper. In this method, the stepped frequency offset of FDA is introduced to produce multiple barrage patches along the range direction, and the size and the distribution of barrage patches are analyzed. Few of the suggestions are:
1. There must be a comparison table inlcuded at the end of the paper to compare the key achievements of the proposed work with the literature work.
2. All the results seem to be based on mathematical modelling or simulation. Can authors demonstrate some experimental verfication as well of the proposed concept?
3. It would be nice, if authors can specify some specific applications of the proposed work or it can be only used in radar applications.
Author Response
Thanks a lot for your comments and those kind suggestions of our manuscript entitled "Two-dimensional barrage jamming against SAR using Frequency diverse array jammer". We provide this cover letter to explain, point by point, the details of our revisions in the manuscript and our responses to the comments as follows. Besides, we have carefully checked through the whole manuscript and corrected all the mistakes found. In order to make the changes easily viewable for you and the reviewers, in our revised paper, we highlighted the revisions with yellow color.
Point 1: There must be a comparison table included at the end of the paper to compare the key achievements of the proposed work with the literature work.
Response 1: Thank you for the comment. In our manuscript, the literature works mainly include the direct repeater jamming with fixed frequency offset using FDA jammer, the deceptive jamming using FDA jammer. Firstly, since the proposed jamming method is mainly for the generation of barrage jamming, it is not compared with the deceptive jamming using FDA jammer. Secondly, compared to the direct repeater jamming with fixed frequency offset, the improvements of the proposed method are mainly in coverage of jamming, diversity of jamming effects, etc., which are mostly qualitative performance improvements and less suitable to be illustrated by a comparison table. In order to give the reader a better understanding of the superiority of the proposed method, we have included a description of its comparison with the direct repeater jamming with fixed frequency offset in the revision manuscript. The corresponding revisions are on Page 20, Line 496-501, and we paste it here for your check. To sum up, in contrast to the direct repeater jamming with fixed frequency offset which can only generate point false targets distributed along the range direction, the proposed method creates multiple barrage patches in the range direction by introducing the stepped frequency offset, and extends the coverage of the barrage patches to the azimuth direction by combining micro-motion modulation, resulting in a 2-D barrage jamming with variable effects.
Point 2: All the results seem to be based on mathematical modelling or simulation. Can authors demonstrate some experimental verification as well of the proposed concept?
Response 2: Thank you for the comment. Unfortunately, we do not have an FDA jammer or a real SAR system at our research facility, so we have not been able to verify the validity of the method through real-measured experiments. Of course, the next phase of our work will be to develop an FDA jammer and conduct real-measured experiments.
Point 3: It would be nice, if authors can specify some specific applications of the proposed work or it can be only used in radar applications.
Response 3: Thank you for you kind advice. As this manuscript is a study of a 2-D barrage jamming method against SAR, which is very specialized, we think it can only be applied to the radar applications at the moment. Of course, we will think further later on whether it can be applied to other applications, and if so, it will greatly increase the value of our research.

Round 2
Reviewer 2 Report
The authors have well addressed all my comments.
Reviewer 3 Report
The revision seems good.